# Initial Study of Fungal Bioconversion of *guishe* (*Agave lechuguilla* Residue) Juice for Bioherbicide Activity on Model Seeds

José Humberto Sánchez Robles [1], Cristina Fernanda Luna Enríquez [1], Ana G. Reyes [2,*],
Marisol Cruz Requena [3], Leopoldo J. Ríos González [1,*], Thelma K. Morales Martínez [1],
Juan A. Ascacio Valdés [4] and Miguel A. Medina Morales [1,*]

[1] Departamento de Biotecnología, Facultad de Ciencias Químicas, Universidad Autónoma de Coahuila, Saltillo 25280, Coahuila, Mexico; jose.sanchez@uadec.edu.mx (J.H.S.R.)
[2] CONACYT—Centro de Investigaciones Biológicas del Noroeste, La Paz 23096, Baja California Sur, Mexico
[3] Departamento de Parasitología, Universidad Autónoma Agraria Antonio Narro, Buena Vista, Saltillo 25315, Coahuila, Mexico
[4] Departamento de Investigación en Alimentos, Facultad de Ciencias Químicas, Universidad Autónoma de Coahuila, Saltillo 25280, Coahuila, Mexico
* Correspondence: agalvarado@cibnor.mx (A.G.R.); leopoldo.rios@uadec.edu.mx (L.J.R.G.); miguel.medina@uadec.edu.mx (M.A.M.M.)

**Abstract:** In agriculture, weed management is a significant concern because their uncontrolled proliferation decreases soil quality for food crops. Allelopathy is a natural phenomenon in which the activity of allelochemical compounds inhibits the germination and growth of invasive plants as a defense mechanism. Among allelochemicals are polyphenols, which may affect genetic material or crucial enzyme activities for proper physiological function. Agroindustrial residues are a vast source of polyphenolic compounds with allelochemical activity. The bagasse of Agave Lechuguilla, known as *guishe*, is an abundant residue in México. The *guishe* has been characterized before by its polyphenolic content. Based on that, a fungal bioconversion process was developed to increase the availability of the allelochemicals in the *guishe* juice. First, *guishe* juice was obtained by mechanical pressed and characterized by spectrophotometric analysis. Results showed (g/L): 5.62 flavonoids, 0.64 of hydrolyzable polyphenols, 12.67 of reducing sugars, and 23.3 total sugars. The compounds detected by HPLC-ESI-MS were pterostilbene, hydroxycaffeic, caffeoyltartaric, and 4-O-glucoside coumaric acids, considered allelopathic. After the fungal bioprocess, (+)-gallocatechin and 3,7-Dimethyl quercetin were detected as additional compounds of interest. The flavonoid and hydrolyzable polyphenol content were modified to the highest accumulation of 1.57 and 14.9 g/L at 72 h, meaning a 2.45- and 2.22-fold increase. A bioprocess *guishe* juice (BGJ) was obtained at the compound accumulation peak of 72 h and evaluated in an allelopathic assay on model seeds (tomato and corn). Results show that BGJ inhibits up to 96.67% of corn seeds and up to 76.6% of tomato seeds compared to positive control.

**Keywords:** bioprocess; allelopathy; bioherbicide; bioconversion; Agave; residues

## 1. Introduction

Weed management is an area in agriculture that holds great importance because the uncontrolled proliferation of weeds can lower soil quality for crops, reducing nutrients, nitrogen, and water and hindering crop growth [1]. The losses caused by weeds in agriculture surpass any other agronomic pests. They reduce plant development and grain production and can mean a 20 to 40% loss in crop yield [2,3]. In this regard, agriculture needs herbicidal agents to alleviate or inhibit weed growth because it causes affection in crops and significant economic losses [4]. The herbicidal agents of synthetic origins, such as glyphosate, have been classified as potentially carcinogenic along with metabolic

and endocrine effects, tumorigenic, teratogenic effects, and hepatorenal damage [5]. The approaches of mechanic removal and cultural management have proved ineffective, and the synthetic options are effective but are considered prejudicial for the environment and human health [6].

Interestingly, allelopathy is a natural phenomenon in which chemicals are released from one plant to the environment to affect neighboring plants and cause hindering effects in germination, growth, and development [7]. The compounds are responsible for allelopathy are known as allelochemicals, in which phenolic compounds are considered [8]. Molecules such as phenolic acids, and flavonoids including flavones, flavanones, isoflavones, and flavonols are reported as weed inhibitors and are also considered environmentally friendly [7,8]. Natural sources of phenolic compounds can be different plants, flowers, fruits, and vegetables [9]. Although, the use of agroindustrial byproducts or vegetable residues is a more convenient alternative for extracting phenolic compounds with allelochemical activity due to their abundance and their attractiveness in economically feasible and green production strategies [8]

*Agave lechuguilla* is a succulent plant found in the semi-desertic area of Mexico and is widely distributed in the country's total area [10]. Rural farmers use it to extract fibers as a productive activity. During the obtention of *A. lechuguilla* fibers, an abundant residue called *guishe* is generated [11]. Several bioactive molecules can be found in this residue, such as saponins, phenolic acids, and flavonoids [12–14]. Among these molecules, some with the ability to inhibit weed growth can be found [15]. The allelopathy these compounds show opens the possibility for *guishe*-based bioherbicides development. Although, their use as a raw material for bioactive compound extraction represents challenges to overcoming the low bioavailability of the molecules [16]. Therefore, biotechnological processes can be adapted to increase the allelochemicals' concentration and bioavailability in the *guishe*. In this regard, some microorganisms can act as agents to release bioactive compounds linked to larger molecules, such as polysaccharides or organic sources containing those molecules, which can be allelochemicals [17]. Considering this, fungal fermentation is a biotechnological approach by which it is possible to potentiate the allelopathy, promoting allelochemical accumulation.

Fungi help achieve the bioaccumulation of bioactive compounds. Fungal strains have been used to treat plant material with phenolic compounds to increase their effects, such as antioxidant activity. Consequently, they may produce enzymes to degrade plant components as a carbon and nitrogen source [18]. Strains such as *Fusarium chlamydosporum* have been used for their ability to accumulate bioactive phenolics in recent reports [19]. Fermented materials with fungal strains have improved their antioxidant capacity after being released by fermentation. In this case, it was mentioned that the bioavailable compounds might inhibit enzymes such as amylase, which is crucial in seed starch degradation [20]. Accordingly, this work aimed to valorize the *guishe* juice by its fungal bioconversion to accumulate allelochemicals using an endophytic strain. In addition, to determine the bioconverted extract capacity to prevent model seed growth, its allelopathic ability was evaluated on *Zea mays* and *Solanum lycopersicum* seeds.

## 2. Materials and Methods

### 2.1. Plant Extract Conditioning

*Agave lechuguilla* residue (*guishe*) was obtained from the Fraustro Coahuila community (25°52′11″ N, 101°19′50″ W). The *guishe* juice was obtained from the whole lechuguilla residue after carving for fiber obtention by hydraulic pressing. The obtained *guishe* juice was stored in a freezer at −20 °C until further use for characterization and evaluations.

### 2.2. Fungal Strain Growth and Fermentation Conditions

The fungal strain of *Fusarium chlamydosporum*, isolated from *Agave lechuguilla* residues, was grown in potato dextrose agar (PDA) from glycerol-cryopreserved spores stored at −20 °C. The agar plates were incubated at 30 °C for 5 days. A spore count was made in a

Neubauer chamber to determine the number of spores harvested with 0.1% Tween 80. The inoculum established was $1 \times 10^6$ spores/mL of *guishe* juice. For the fermentation conditions, in 250 mL Erlenmeyer flasks, 50 mL of undiluted *guishe* juice was the culture media and stirred at 150 rpm. The *guishe* juice was previously centrifuged at 6000 rpm/10 min to remove solid particles. The fermentation was carried out for 168 h, with samples taken every 24 h. The samples of BGJ were centrifuged and stored in a $-20\,^{\circ}$C freezer until analysis.

### 2.3. Analytical Methods

2.3.1. Spectrophotometric Analysis

For flavonoid content, the samples were analyzed by the technique reported by Silva-Beltrán et al. [21]. For hydrolyzable phenolics (HP), the method carried out was reported by Wong Paz et al. [22]. Total sugars were determined by the phenol-sulfuric method [23], and reducing sugars by the dinitrosalicylic acid method [24]. Results from experiments were made in triplicate, and calibration curves were prepared to quantify the equivalents of quercetin, gallic acid, and glucose, respectively.

2.3.2. Phenolic Content by HPLC-ESI-MS

The bioactive molecules were detected by sample injection in high-performance liquid chromatography (Varian ProStar), as reported by Estrada-Gil et al. [25]. The samples were filtered through a 0.45 μm nylon membrane with a 1:10 dilution. The equipment uses a ternary pump (ProStar 2301), an autosampler (ProStar 410), a photodiode array detector (ProStar 330) (320 nm), and a Denali C18 column (3.1 μm; 150 mm $\times$ 4.6 mm) at 30 $^{\circ}$C in a column oven. The mobile phases used were 0.2% formic acid as solvent A and acetonitrile as solvent B. The solvents were used by gradients as follows: initial 3% B; 5–15 min, 16% B linear; and 15–45 min, 50% B linear. After the analysis, the column was washed and reconditioned. The chromatographer was coupled with a mass spectrometer with ion trap (Varian 500 M/S), electrospray ionization (ESI), negative mode [M-H]-, 90 V of capillary voltage, and 100–2000 $m/z$ of mass range. Full scan mode acquired in the $m/z$ range of 50–2000 was the analysis made on the samples, and the data collected were processed by the MS Workstation Software (version 6.9).

### 2.4. Germination and Vegetative Growth Effects

Model seeds were used to evaluate the effect of the bioprocess *guishe* juice (BGJ) under laboratory conditions. The seeds were maize (*Zea mays*) and tomato (*Solanum lycopersicum*) as models. Seeds were disinfected with 1% NaOCl for 15 min, washed with distilled water, and dried. The seeds were placed in Petri dishes with humidified paper filter discs. As controls, the seeds were imbibed with water (negative control) and glyphosate-based product (positive control). Later the seeds were imbibed in the bioprocess *guishe* juice (BGJ) for 15 min, where the extract was evaluated in four conditions: undiluted, 2%, 1%, and 0.5%. The germination index and vegetative growth were assessed for 3 days by observing the emerging roots' development and characteristics in a climatic chamber at 80% moisture, 25 $^{\circ}$C ($\pm 2.5\,^{\circ}$C) in complete darkness.

### 2.5. Data Analyses

All experiments were carried out in triplicate, and results were analyzed in Microsoft Excel for means, standard deviations, and graphics. Analysis of variance and comparison of means by Tukey's test (at 0.05) using Minitab$^{®}$ 20.3 (64-bit) statistical software was used for the seed germination experiment results.

## 3. Results

### 3.1. Guishe Juice Characterization

Table 1 shows the flavonoid, hydrolyzable phenolics, reducing and total sugar content detected by spectrophotometric methods. The sugar content was 23.3 g/L by the sulfuric-

phenol method and 12.67 g/L by the DNS method as glucose equivalents. For the phenolics and flavonoid content, 0.64 and 5.62 g/L, were detected as quercetin and gallic acid equivalents, respectively.

**Table 1.** Spectrophotometric characterization of *guishe* juice before the fungal fermentation process.

| Compounds of Interest | g/L |
|---|---|
| Total sugars | 23.3 ± 0.90 |
| Reducing sugars | 12.67 ± 0.170 |
| Hydrolyzable polyphenols | 0.64 ± 0.014 |
| Flavonoids | 5.62 ± 0.024 |

High-performance liquid chromatography detected several bioactive compounds, as seen in Table 2. Compounds such as pterostilbene, hydroxycaffeic acid, caffeoyltartaric acid, and 4-O-glucoside coumaric acid were detected. The separation was made with a gradient method and a C-18 column (specified in the methodology section) with an affinity for polyphenols and their derivates.

**Table 2.** Compounds detected in *guishe* juice characterization by HPLC-ESI-MS.

| Mass (*m/z*) | Compound | Family | [1] RT (min) |
|---|---|---|---|
| 254.9 | Pterostilbene | Stilbenes | 14.96 |
| 194.9 | Hydroxycaffeic acid | Hydroxycinnamic acids | 21.183 |
| 325 | 4-*O*-glucoside *p*-coumaric acid | Hydroxycinnamic acids | 54.594 |
| 311.1 | Caffeoyltartaric acid | Hydroxycinnamic acids | 55.685 |

[1] Retention time.

### 3.2. Bioprocessing of Guishe Juice

Sugar content acts as a growth inducer along the fungal bioprocessing, and its content is affected through time, as seen in Figure 1. The bioprocess started with a sugar level of 23.03 g/L, with a decrease in content levels to 8.18 g/L at 48 h. Afterward, a tendency to accumulate sugars was observed at 96 h with 13.5 g/L, and the lowest sugar level was detected at 5.3 g/L. Approximately 12 g/L of sugars was maintained until 168 h during the final three monitoring times. In Figure 2, the reducing sugar content is shown. At the beginning of the bioprocess, the content of monomeric sugars was 12.6 g/L, and as time progressed, sugar accumulation was observed until 120 h. Between 24 and 72 h, an approximate amount of 23 g/L was maintained. Afterward, an irregular behavior in sugar content was registered at 120 h until 168 h.

### 3.3. Flavonoids and Hydrolyzable Phenolics

In Figure 3, the THP content is shown. At the beginning of the bioprocess, the THP quantity detected was 0.64 g/L; as time advanced, 1.57 g/L of THP was accumulated at 72 h, and 1.29 g/L of THP at 168 h was detected. For flavonoid content, the presence of the compound evolved from 5.62 g/L to an accumulation of 14.9 g/L at 72 h to a final concentration of also 14.9 at 168 h, as seen in Figure 4. In both cases, polyphenols show an accumulation pattern which indicates an active polyphenol accumulation.

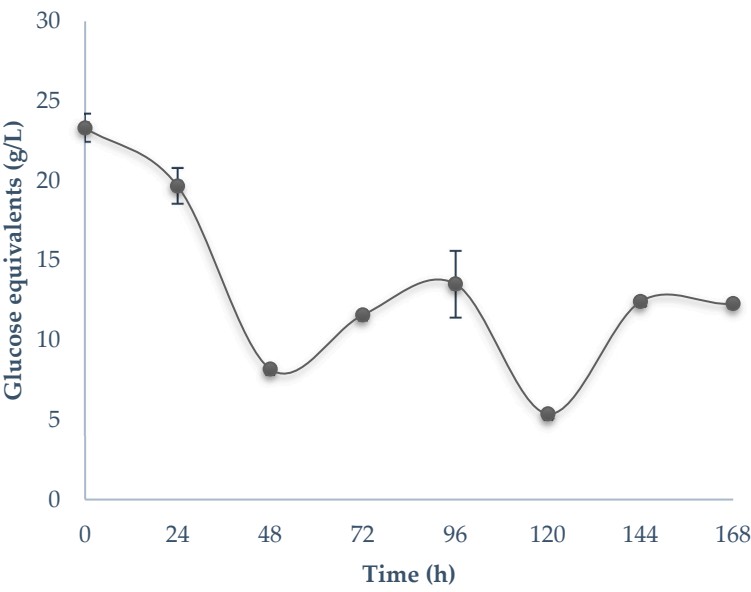

**Figure 1.** Total sugars content along the fungal bioprocess of *guishe* juice.

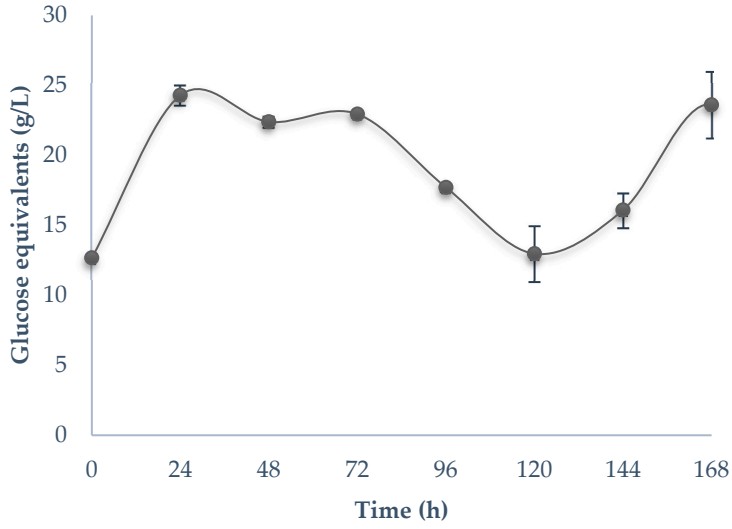

**Figure 2.** Reducing sugars content along the *Fusarium chlamydosporum* bioprocess *guishe* juice (BGJ).

In Table 3, the compounds detected in the bioprocess *guishe* juice (BGJ) are shown by HPLC-ESI-MS. The difference with the characterized *guishe* juice is that other compounds were detected at 24 h and afterward, such as (+)-gallocatechin and 3,7-dimethyl quercetin bioactive compounds. The only glycoside found was 4-o-glucoside p-coumaric acid, and its presence was maintained for 72 h. (+)-Gallocatechin was a compound that appeared at 24 h and prevailed until 96 h, and this molecule may have been part of a complex that was degraded by enzymes secreted by the fungal strain. Hydroxycaffeic acid and pterostilbene were detected during the whole bioprocess. The chemical structures of the detected compounds by HPLC-ESI-MS are shown in Figure 5.

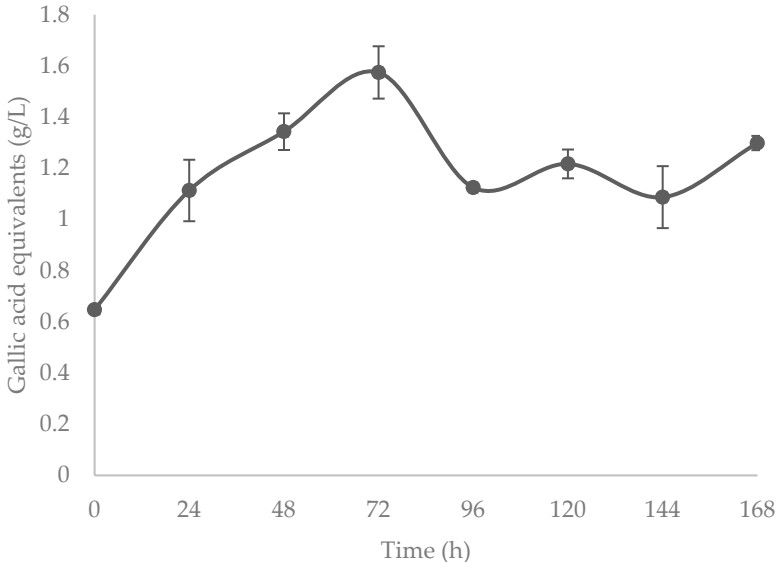

**Figure 3.** Total hydrolyzable phenolics content in the bioprocess *guishe* juice (BGJ).

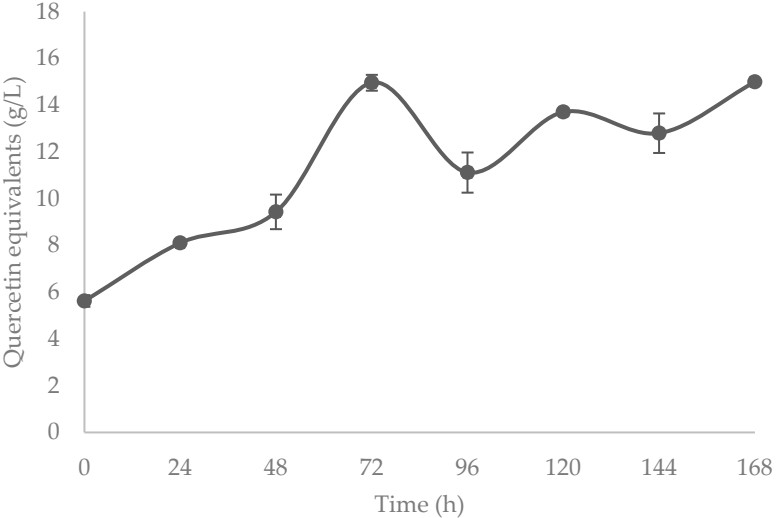

**Figure 4.** Flavonoid content in the *Fusarium chlamydosporum* bioprocess *guishe* juice (BGJ).

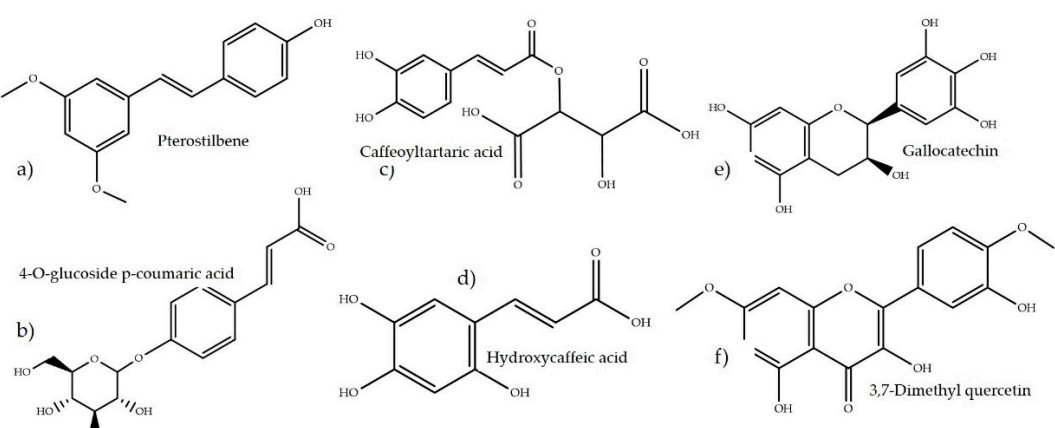

**Figure 5.** Bioactive compounds detected in plant material. Compounds from (**a–d**) were detected in the unprocessed *guishe* juice. Compounds in (**e,f**) were detected in the bioprocess *guishe* juice BGJ, as seen in Tables 2 and 3.

**Table 3.** Compounds detected by HPLC-ESI-MS along the BGJ with *Fusarium chlamydosporum*.

| Mass (*m/z*) | Compound | Family | [1] RT (min) |
|---|---|---|---|
| | 24 h | | |
| 304.7 | (+)-Gallocatechin | Catechins | 3.466 |
| 254.9 | Pterostilbene | Stilbenes | 14.77 |
| 195 | Hydroxycaffeic acid | Hydroxycinnamic acids | 20.542 |
| 917 | 3,7-Dimethyl quercetin | Methoxyflavones | 55.706 |
| 325 | 4-O-glucoside p-coumaric acid | Hydroxycinnamic acids | |
| | 48 h | | |
| 304.7 | (+)-Gallocatechin | Catechins | 3.336 |
| 254.9 | Pterostilbene | Stilbenes | 14.859 |
| 194.9 | Hydroxycaffeic acid | Hydroxycinnamic acids | 21.513 |
| 325 | 4-O-glucoside p-coumaric acid | Hydroxycinnamic acids | 55.948 |
| | 72 h | | |
| 304.7 | (+)-Gallocatechin | Catechins | 3.39 |
| 254.9 | Pterostilbene | Stilbenes | 14.249 |
| 194.9 | Hydroxycaffeic acid | Hydroxycinnamic acids | 20.825 |
| 325 | 4-O-glucoside p-coumaric acid | Hydroxycinnamic acids | 55.73 |
| | 96 h | | |
| 304.7 | (+)-Gallocatechin | Catechins | 3.504 |
| 254.9 | Pterostilbene | Stilbenes | 14.515 |
| 195 | Hydroxycaffeic acid | Hydroxycinnamic acids | 21.071 |

[1] Retention time.

### 3.4. Allelopathic Activity

In the assays carried out on *Zea mays* seeds (Table 4), it was observed that with the addition of 100% bioprocess guishe juice (BGJ), germination was inhibited at 100% at the first 24 h of incubation followed by 48 and 72 h with 96.67%, respectively, compared with the control. At a lower concentration of 0.5% of bioprocess extract, the germination inhibition was 60% at 24 h of incubation, and later, results showed 3.33% and 0% at 48 and 72 h, respectively. A similar effect was observed at 1 and 2%, the concentration of the BGJ, compared with the control. For this analysis, water and glyphosate were controls; water control treatment displayed normal root formation after 72 h. The positive control had a 70% germination, but the roots were affected considerably compared to the negative control. As seen in Figure 6, the seeds developed roots that also have differences. In the water control (Figure 6a), the roots are thicker than in the glyphosate control (Figure 6b) and 0.5 % BGJ treatments (Figure 6c), and in 100% BGJ, no root formation was observed (Figure 6d). By statistical analysis, it can be corroborated that 100% BGJ is significantly different compared to the other treatments. At 72 h, the effect is even more notorious, considering that inhibition was already manifesting at 24 and 48 h.

**Table 4.** Allelopathic activity on *Zea mays* model seeds treated with BGJ fermented during 72 h, glyphosate as the positive control, and water as the negative control.

| Treatment | 24 h | | 48 h | | 72 h | |
|---|---|---|---|---|---|---|
| | | | Control treatment | | | |
| Water | 70 | abc | 0 | a | 0 | b |
| Glyphosate | 70 ± 5.77 | ab | 0 | a | 0 | b |
| | | | Bioprocesses *guishe* juice | | | |
| 0.50% | 60 ± 17.32 | bc | 3.33 ± 5.77 | a | 0.00 | b |
| 1.00% | 36.67 ± 20.82 | c | 3.33 ± 5.77 | a | 0.00 | b |
| 2.00% | 36.67 ± 15.28 | c | 3.33 ± 5.77 | a | 0.00 | b |
| 100% | 100 | a | 96.67 ± 5.77 | a | 96.67 ± 5.77 | a |

Means difference is significant at the 0.05 level. Means that do not share a letter are significantly different.

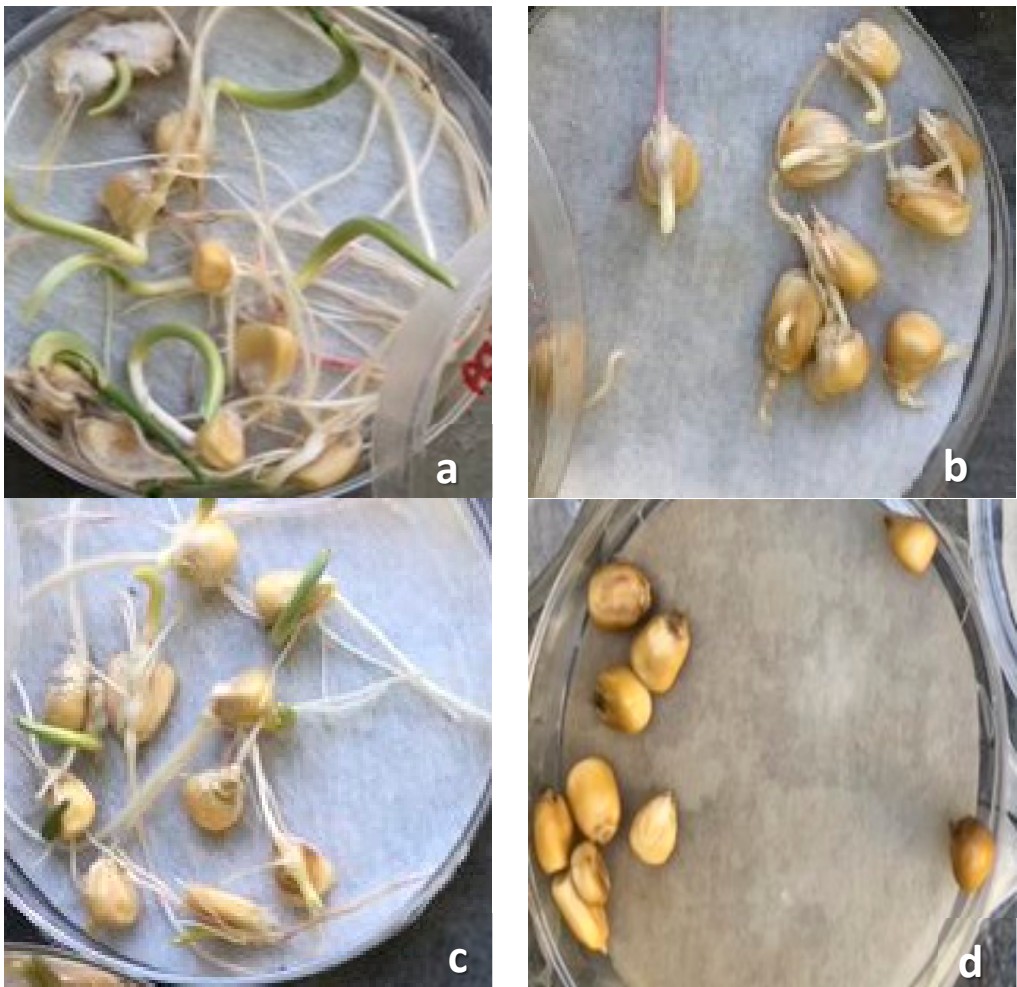

**Figure 6.** Germination inhibition and root formation in treated seeds of *Zea mays* where (**a**) is water control treatment, (**b**) is glyphosate, (**c**) is 0.5% BGJ, and (**d**) is 100% BGJ.

For *Solanum lycopersicum* seeds (Table 5), at the first 24 h of all BGJ treatments, seeds did not germinate. After that, at 48 h of incubation, 86% of the seeds were inhibited at the 100% concentration of BGJ, and 76% of the seeds were inhibited at 72 h. The activity decreased at lower concentrations of BGJ, where at 0.5% of BGJ at 48 h, 33.33% of seeds were inhibited, and 26.67% at 72 h. For 1% BGJ at 48 and 72 h, the inhibition resulted in 32 and 26.67%, and at 2% concentration of BGJ, the inhibition at 48 and 72 h was 30 and 16.67%. The glyphosate inhibited 63.33, 36.67, and 33.33% of the seeds from 24 to 72 h, and water inhibited 30, 53.33, and 53.33 in the same periods, where in this case, it may be possible that the seeds in the water treatment had low viability as all seeds were expected to germinate in its totality. In Figure 7, tomato seeds also exhibited typical water treatment growth in the negative control (water), decreased root volume in the positive control treatment (glyphosate), and a near-total inhibition of root formation in 100% BGJ. The effect of 100% BGJ was significantly different for tomato seeds compared to controls and diluted BGJ. The allelopathic effect was detected for 48 h of growth, which became more evident at 72 h. Water and glyphosate were used as control treatments.

**Table 5.** Allelopathic activity on *Solanum lycopersicum* model seeds treated with BGJ fermented during 72 h, glyphosate as the positive control, and water as the negative control.

| Treatment | 24 h | | 48 h | | 72 h | |
|---|---|---|---|---|---|---|
| | | | Control treatment | | | |
| Water | 30 | a | 46.67 ± 5.77 | abc | 46.67 ± 5.77 | ab |
| Glyphosate | 63.33 | a | 63.33 ± 5.77 | ab | 66.67 ± 5.77 | ab |
| | | | Bioprocess *guishe* juice | | | |
| 0.50% | 100 | a | 33.33 ± 15.28 | bc | 26.67 ± 20.82 | b |
| 1.00% | 100 | a | 32.00 ± 11.55 | bc | 26.67 ± 15.28 | b |
| 2.00% | 100 | a | 30.00 ± 20.00 | c | 16.67 ± 15.28 | b |
| 100% | 100 | a | 86.67 ± 11.55 | a | 76.67 ± 11.55 | a |

Means difference is significant at the 0.05 level. Means that do not share a letter are significantly different.

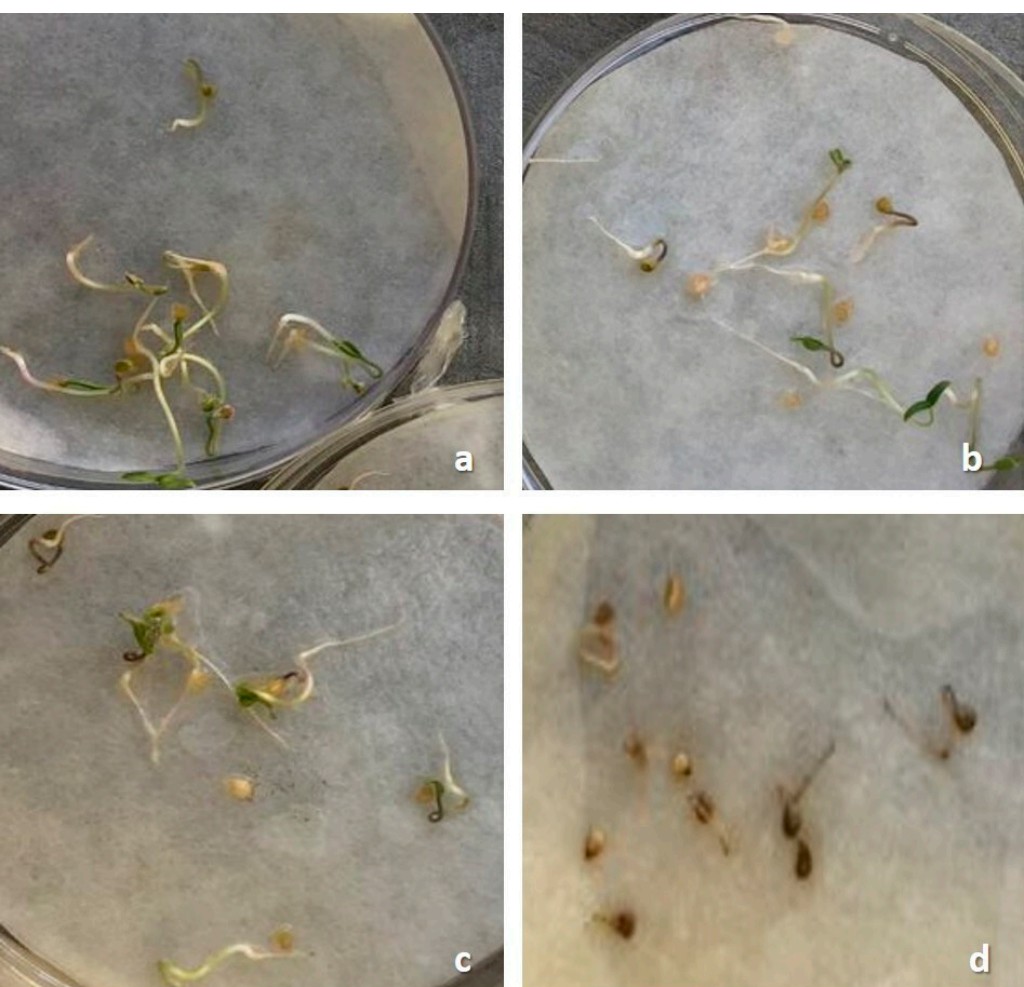

**Figure 7.** Root formation and germination inhibition in treated seeds of *Solanum lycopersicum* seeds. (**a**) is water control treatment, (**b**) is glyphosate, (**c**) is 0.5% BGJ, and (**d**) is 100% BGJ.

## 4. Discussion

### 4.1. Guishe Juice Characterization

The sugar content can be attributed to fructans and their derivates, considering that *Agave* species can accumulate these sugars as energy sources and microbial growth inducers [11,12]. By liquid chromatography coupled with mass spectroscopy, several bioactive compounds were detected, as seen in Table 2. The separation was made in a method and a column with an affinity for polyphenols and their derivates and compounds, such as

pterostilbene, hydroxycaffeic acid, caffeoyl-tartaric acid, and 4-*O*-glucoside coumaric acid. These compounds are among flavonoids and stilbenes, which are reported to manifest bioactivities, and in the case of this work, one of such bioactivities is allelopathy [5,26]. Caffeic acid has been reported as a down regulator of the genes for the biosynthesis of phytoalexin and gibberellins, which are compounds that contribute to seed germination and stem development [27]. The effect of p-coumaric acid is mentioned as a stress factor in plant development due to its effect as a photosynthesis inhibitor [28]. Stilbenes, including pterostilbene, are molecules studied to assess their allelopathic activity against the genus *Fallopia*, considered one of the worst invasive species in the world [29]. Gallocatechin and quercetin can affect seed germination and nitrification processes [4]. Many compounds, reported as allelopathic, can express bioactivity even in derivates such as glycosides [8].

*4.2. Bioprocessing of Guishe*

In Figure 2, the reducing or monomeric sugars accumulation can be explained by the hydrolysis of probable fructooligosaccharides and inulins and their later uptake by the fungal strain [30]. This behavior in sugar content could be attributed to the heterogeneity of the substrate, guishe juice, in this case. Some reports mention that guishe contains monomeric sugars such as glucose, fructose, fructooligosaccharides, and inulins. Some fungi can produce enzymes to degrade fructooligosaccharides. This degradation can contribute to the sugar accumulations observed in Figure 3 [31]. Sugar content and other molecules (glycosides) contained within the extract may induce *Fusarium chlamydosporum* to secrete hydrolytic enzymes, which may contribute to accumulating bioactive compounds and increase their bioavailability. As sugars were consumed, other compounds, such as polyphenols, were accumulated, as seen in Figures 3 and 4. This effect can promote increased bioavailability of the compounds towards the desired allelopathic activity. By decreasing sugar content, the possibilities of other bioactivities inherent in plants are lower, such as defense mechanisms associated with plant immunity and different molecular patterns suggesting that carbohydrates of diverse molecular weights or polymerization degrees may activate plant signaling cascades [32].

*4.3. Flavonoids and Hydrolyzable Phenolics*

Bioactive phenolic compounds can be found in plant tissues with glycosylation in their structures, which allows water solubility compared with the aglycone form of insoluble phenolics [33]. The accumulation patterns among phenolic compounds may follow the enzymes' activity to hydrolyze certain compounds and the uptake of sugar content by the fungal strain. It can be inferred that the compounds in the media are being degraded by the active bioprocess and by a wide range of enzymes a fungus can produce. It contributes to the bioavailability of molecules [34].

Fungal strains have been used in fermentation systems where polyphenols were the molecules of interest. Hassane et al. [19] attributed the biodegradation ability of *Fusarium chlamydosporum* on humidified rice to the release of flavonoids such as apigenin, chrysin, naringenin, isovitexin, and quercetin. Medina-Morales et al. [35] reported a mixed culture of *Aspergillus niger* and *Trichoderma reesei* to study the release of polyphenolic compounds bound to fibers in citric residues where enzymatic activities such as β-glucosidase, cellulase, and xylanase stood out. There are studies where fungal strains can proliferate and degrade polyphenolic compounds. López Trujillo et al. [34] and Xue et al. [36] reported that *Aspergillus niger* could accumulate flavonoids such as luteolin and apigenin glycosides. Some reports mention that glycosidic enzymes such as b-glucosidases can degrade the bond in a phenolic compound and a sugar to release an aglycone and a monosaccharide [37]. Another type of catalytic activity is oxygenases, which can act directly at the C-C bonds in flavonoids and may contribute to the release of phenolic compounds [38], allowing the accumulation seen in Figures 3 and 4. According to Morreeuw et al. [14], these compounds can be found in guishe considering that we worked with a mechanically obtained extract and the mentioned report used whole guishe, which are water-soluble compounds. As

previously mentioned, all these compounds and their derivates have been reported to have bioactivity, including allelopathy. These aspects are relevant because plants can express a defense mechanism that hinders invasive plant growth in their proximity. Because of these reasons, natural phenolic compounds are beneficial to crops for weed management, as these can be found in aqueous extracts containing these molecules [8].

### 4.4. Allelopathic Activity

Several mechanisms affect the germination and growth of plants through allelopathy, such as cellular effects on division and permeability, oxidative stress, photosynthesis, respiration, transpiration, gene expression, protein biosynthesis, phytohormones, and enzyme activities [8,39]. The results show that in *Zea mays* seeds, the roots and stems, according to Figure 6, may have been affected by the compounds present in the BGJ. Nadeem et al. [40] mentioned that phenolic compounds found in *Alternanthera philoxeroides* had a negative development effect on the roots and stems of *Zea mays* seeds and attributed this to ferulic acid, chlorogenic acid, gallic acid, p-coumaric acid, syringic acid, and quercetin. The results of tomato seeds (Figure 7) show that they were slightly more resistant to the BGJ, which could be attributed to natural resistance. It is reported that tomato roots are in contact with allelochemicals in the soil, which could give resistance [41]. The differences in the root formation in the treated and untreated seeds may affect structure development and protein synthesis that decrease the width and length of the roots, resulting in a weaker seedling and less likelihood to continue its growth. One of the reasons that phenolic compounds can be found in glycoside form, apart from improving their transit in and out of plant tissues, is to avoid autotoxicity [2], so in our results, phenolic glycosides were found and adjusted to the observed allelopathic activity to its initial expectations. As previously stated, the evaluation of BGJ at the highest concentration presented total or near total inhibition of the model seeds. The presence of phenolic compounds may have a role in this regard due to their effect on cell permeability, photosynthesis, and nucleic acid synthesis, which alters gene expression and metabolism, such as phytohormones and regulators in plant systems. Phenolic compounds are among the first line of defense for plants considered allelopathic, or the compounds themselves are allelopathic agents, regardless of their origin. One example is that caffeic acids inhibit phosphorylation and its energetic processes in seedlings. Cinnamic acids can tamper ATPase production, lowering nitrogen uptake and affecting growth [39,42].

Since glycosides may have reduced allelopathic activity, the partial biodegradation by *F. chlamydosporum* could contribute to an increased germination inhibition due to the glycosidases required to release the phenolic compounds. There are reports that glycosides can be degraded to release aglycones [34,43], which may be phenolic compounds (phenolic acids and flavonoids), which penetrate the seed and inhibit enzymes activities responsible for germination, among other mechanisms [5,42].

### 5. Conclusions

This is the first study demonstrating that the bioconversion of *guishe* juice with *Agave lechuguilla* endophytic *Fusarium chlamydosporum* fermentation is an attractive biotechnological strategy. The fungal bioconversion of guishe juice (BGJ) promotes the accumulation of phenolic and flavonoid compounds, known for their allelopathic activities. Bioconversion is useful for accumulated compounds like phenolic acids and flavonoids such as gallocatechin and glycosylated derivatives. The accumulation of these compounds allows the development of guishe-based herbicides. This work showed the allelopathic effect of BGJ on model seeds.

**Author Contributions:** Conceptualization, M.A.M.M. and J.H.S.R.; Methodology, C.F.L.E., J.H.S.R., M.C.R., A.G.R., T.K.M.M. and J.A.A.V.; Software, J.H.S.R.; Validation, M.A.M.M., L.J.R.G. and A.G.R.; Formal Analysis, M.A.M.M.; Investigation, L.J.R.G.; Resources, L.J.R.G., M.A.M.M. and A.G.R.; Data Curation, T.K.M.M., J.A.A.V. and J.H.S.R.; Writing—Original Draft Preparation, J.H.S.R. and M.A.M.M.; Writing—Review and Editing, All authors; Supervision, M.A.M.M., L.J.R.G. and A.G.R.;

Project Administration, M.A.M.M. and L.J.R.G.; Funding Acquisition M.A.M.M., L.J.R.G. and A.G.R. All authors have read and agreed to the published version of the manuscript.

**Funding:** This research was funded by the National Council of Science and Technology (CONACYT) by granting scholarships to students, and for the funded project 322622 "Optimización y validación de la efectividad de un bioherbicida formulado a base de plantas de uso tradicional del semidesierto mexicano". Partially supported with materials and reagents from funded project PN-2017-7332 by CONACYT.

**Institutional Review Board Statement:** Not applicable.

**Informed Consent Statement:** Not applicable.

**Data Availability Statement:** The data presented in this study are available on request from the corresponding author.

**Acknowledgments:** We would like to thank the National Council of Science and Technology for awarding a research grant (322622) which included this research.

**Conflicts of Interest:** The authors declare no conflict of interest.

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
