# Peer review of "Initial Study of Fungal Bioconversion of guishe (Agave lechuguilla Residue) Juice for Bioherbicide Activity on Model Seeds"

_fermentation, doi:10.3390/fermentation9050421_

Round 1
Reviewer 1 Report
In this work, a fungal bioconversion process to increase the availability of the allelochemicals in agave guishe juice was developed. The experiment design was simple. Most Importantly, the language and logic of this manuscript are strongly suggested to be polished. Other comments are listed as follows.
1 The name of Fusarium chlamydosporum in the manuscript should be written in italic.
2 In the line 93, " Total sugars were carried out by" should be shown as " Total sugars were detected (or tested or determined) by".
3 In the line of 97, which molecules were detected, therefore, the authors should clearly indicate the names of these determined molecules.
4 In the lines of 115 to 116, pls check this sentence "The germination index and vegetative growth were assessed for 3 days by observing germination".
5 In the line of 123, pls revise this sentence "0.64 and 5.62 g/L were detected,".
6 In the line of 127, the authors should clearly demonstrate the separation method.
7 In the line of 135, this sentence "with a drop in those levels at 48 hours with 8.18 g/L" is suggested to be revised to "with a drop to 8.18 g/L in the level at 48 hours ".
8 In the lines of 142 to 143, pls revise this sentence "Afterwards, a decrease-accumulation decrease behavior was registered at 120 hours until 168 hours.".
9 I strongly suggest that the sentences in the lines of 148 to 152 should be corrected.
10 In the lines of 177 to 179, the sentence should be revised.
11 In the line of 182, what's the seeds grew roots? pls check it.
12 In the line of 191, pls check this sentence " the 100% concentration of BGE inhibited 86% and 76% at 72 hours".
13 In the line of 227 to 228, pls revise this sentence " Sugar content, along with the other types of molecules contained within the extract,"
14 in the line of 236, "uptake of sugar content", content can be uptaken?
15 In the line of 244, b-glucosidase should be β-glucosidase.
16 Pls revise the sentence in the lines of 264 to 267.
17 The conclusions have no any meaning, it needs to be rewritten.
18 In the manuscript, the authors paid more attentions to simple description on experimental result, being lack of the explanation on the reasons that caused these results.
Author Response
In this work, a fungal bioconversion process to increase the availability of the allelochemicals in agave guishe juice was developed. The experiment design was simple. Most Importantly, the language and logic of this manuscript are strongly suggested to be polished.
-The manuscript was changed so to improve the logic of the text. These aspects, combined with the corrections made, prompted us to improve results and discussions. In the introduction, the sequence of the information provided was changed to match the abstract.
Other comments are listed as follows.
1 The name of Fusarium chlamydosporum in the manuscript should be written in italic.
-The modification has been made
2 In the line 93, " Total sugars were carried out by" should be shown as " Total sugars were detected (or tested or determined) by".
-Modification in methodology has been made: ‘’Total sugars were determined by the phenol-sulfuric method’’ in line 112
3 In the line of 97, which molecules were detected, therefore, the authors should clearly indicate the names of these determined molecules.
-The molecules detected were flavonoids and phenolic acids which are indicated in tables 2 and 3
4 In the lines of 115 to 116, pls check this sentence "The germination index and vegetative growth were assessed for 3 days by observing germination".
-The lines were modified The germination index and vegetative growth were assessed for 3 days by observing the emerging roots' development and characteristics in a climatic chamber at 80% moisture, 25 °C (± 2.5°C) in complete darkness. 137-139
5 In the line of 123, pls revise this sentence "0.64 and 5.62 g/L were detected,".
-The line was revised and modified and now it can be found in lines 150-152
6 In the line of 127, the authors should clearly demonstrate the separation method.
-The separation was made using a gradient method and a C-18 column. This line was added to the text in the corresponding section of chromatographic analysis
7 In the line of 135, this sentence "with a drop in those levels at 48 hours with 8.18 g/L" is suggested to be revised to "with a drop to 8.18 g/L in the level at 48 hours ".
-The line was modified at 165-166
8 In the lines of 142 to 143, pls revise this sentence "Afterwards, a decrease-accumulation decrease behavior was registered at 120 hours until 168 hours.".
-The sentence has been revised and modified and the line “Afterwards, an irregular behavior in sugar content was registered at 120 hours until 168 hours.’’ 172-173
9 I strongly suggest that the sentences in the lines of 148 to 152 should be corrected.
-The lines have been corrected. 180-182
10 In the lines of 177 to 179, the sentence should be revised.
-The lines have been revised and corrected. 209-211
11 In the line of 182, what's the seeds grew roots? pls check it.
-The line was modified to ‘’development’’. 215
12 In the line of 191, pls check this sentence " the 100% concentration of BGE inhibited 86% and 76% at 72 hours".
-The line was modified to be more coherent. 230-231
13 In the line of 227 to 228, pls revise this sentence " Sugar content, along with the other types of molecules contained within the extract,"
-The line has been modified and corrected. 280-281
14 in the line of 236, "uptake of sugar content", content can be uptaken?
-Yes, it may be used as consumption or absorption. If this term does not satisfy the writing or logic of the manuscript, it may be modified.
15 In the line of 244, b-glucosidase should be β-glucosidase.
-Correction made
16 Pls revise the sentence in the lines of 264 to 267.
-This section was enriched with more discusión. 321-338.
17 The conclusions have no any meaning, it needs to be rewritten.
-The conclusions have been adjusted to be more concise.
18 In the manuscript, the authors paid more attentions to simple description on experimental result, being lack of the explanation on the reasons that caused these results.
-We added more information in this regard and it’s colored while other additions go along specific comments made by yourself and another reviewer.
Lines: 242-272, 284-289, 321-338.
Also, the title of the manuscript has been slightly changed and it’s marked in yellow
Reviewer 2 Report
Dear authors,
the manuscript entitled "Initial study of fungal bioconversion of guishe (Agave lechuguilla residue) extract for bioherbicide activity on model seeds" is an interesting idea for the use of post-production waste. The presented conclusions seem to be supported by evidence. However, the manuscript raises some concerns.
Major:
1) Materials and methods in the paragraph: 2.3.1. Spectrophotometric analysis, line 90, should be described in more detail. Taking into account not only the methodology of the analysis, but also information such as the number of experiment repetitions and the number of technical repetitions. The same observation applies to other analyses.
2) 2.3.2. Phenolic content by HPLC-ESI-MS, line 96, - this paragraph lacks information whether standards were used in the analysis or it was based only on the molecular weights of the tested substances. If only molecular weights are used, it is advisable to provide literature data in which test substances were observed using identical analytical techniques.
3) The germination results, like other data, require the use of statistical analysis and, as before, information on the number of experiments performed.
4) Glyphosate as a systemic herbicide seems to be a poor example of a negative control for germination inhibition.
Minor:
Please pay attention to the use of italics for species names: Fusarium chlamydosporum, line 79
I encourage language correction by a native speaker.
With respect,
Reviewer
Author Response
Major:
1) Materials and methods in the paragraph: 2.3.1. Spectrophotometric analysis, line 90, should be described in more detail. Taking into account not only the methodology of the analysis, but also information such as the number of experiment repetitions and the number of technical repetitions. The same observation applies to other analyses.
-The corrections have been made. In all accounts, the analyses were made by triplicate
2) 2.3.2. Phenolic content by HPLC-ESI-MS, line 96, - this paragraph lacks information whether standards were used in the analysis or it was based only on the molecular weights of the tested substances. If only molecular weights are used, it is advisable to provide literature data in which test substances were observed using identical analytical techniques.
-The HPLC-ESI-MS analysis was made without standards because the established method along with the C-18 column, phenolic compounds are the molecules that are separated and detected. The molecular weights and retention times are obtained and the identification is made via a database available in the chromatographer coupled with the mass spectrometer. ‘’For the identification of the molecules, full scan mode acquired in the m/z range of 50-2000 was the analysis made on the samples, and the data collected was processed by the MS Workstation Software (V. 6.9)’’ was the line added at the end of the 2.3.2 section.
3) The germination results, like other data, require the use of statistical analysis and, as before, information on the number of experiments performed.
-The statistical analysis has been included in the form of a Tukey test in the model seeds evaluations results in the tables with the allelopathic results
4) Glyphosate as a systemic herbicide seems to be a poor example of a negative control for germination inhibition.
-Glyphosate was the positive control and we used it as a positive control because it is the central interest of a funded Project granted by our Science Council. Glyphosate is a non-selective systemic herbicide and it’s the most used product (along with its derivates) for weed management in the world which is also why we decided to use it as a positive control
Minor:
Please pay attention to the use of italics for species names: Fusarium chlamydosporum, line 79
-Modification has been made
Also, the title of the manuscript has been slightly changed and it’s marked in yellow
-I encourage language correction by a native speaker.
-We will apply for the english edition services provided by MDPI considering that the text has already been improved in language
-In color blue, more discussion was added and also the introduction was adjusted to the logic from the abstract.

Round 2
Reviewer 1 Report
The authors have now revised this manuscript according to the reviewer's comments. It warrant publication in Fermentation after minor revision.1 "the 100% concentration of BGJ inhibited 86% of the seeds, and 76% of the seeds were inhibited at 72 hours" was corrected to "86% of the seeds were inhibited at the 100% concentration of BGJ, and 76% of the seeds were inhibited at 72 hours"
2 "At the beginning of the bioprocess, the quantity detected was 0.64 g/L; as time advanced, 1.57 g/L was accumulated at 72 hours, and 1.29 g/L at 168 hours was detected" should be revised to "At the beginning of the bioprocess, the THP quantity detected was 0.64 g/L; as time advanced, 1.57 g/L of THP was accumulated at 72 hours, and 1.29 g/L of THP at 168 hours was detected".
3 In the lines 150 to 152, the sentence should corrected as "For the phenolics and flavonoid content, 0.64 and 5.62 g/L, were detected as quercetin and gallic acid equivalents, respectively.". You should remember that it's not the concentration or content to be detected or accumulated, it should be that the matter with a certain concentration is detected or accumulated. As the same as the other similar errors in this manuscript. Pls carefully check and revise this manuscript.
Round 2
RESPONSES TO THE EDITOR´S/REVIEWER’S COMMENTS
Manuscript ID: fermentation-2311375
Title: Initial study of fungal bioconversion of guishe (Agave lechu-guilla residue) extract for bioherbicide activity on model seeds
Dear Editor - Fermentation
Please find below the detailed responses to the Reviewer’s/Editor’s comments. We have also enclosed the thoroughly revised version of the manuscript mentioned above according to the Reviewer’s comments, suggestions, and recommendations. The changes made have been highlighted with a yellow background (Round 1 Revisions) in color as per the Editorial’s recommendation.
Reviewer 1
The authors have now revised this manuscript according to the reviewer's comments. It warrant publication in Fermentation after minor revision.
1 "the 100% concentration of BGJ inhibited 86% of the seeds, and 76% of the seeds were inhibited at 72 hours" was corrected to "86% of the seeds were inhibited at the 100% concentration of BGJ, and 76% of the seeds were inhibited at 72 hours"
- The requested change in the manuscript has been made in the specified lines.
2 "At the beginning of the bioprocess, the quantity detected was 0.64 g/L; as time advanced, 1.57 g/L was accumulated at 72 hours, and 1.29 g/L at 168 hours was detected" should be revised to "At the beginning of the bioprocess, the THP quantity detected was 0.64 g/L; as time advanced, 1.57 g/L of THP was accumulated at 72 hours, and 1.29 g/L of THP at 168 hours was detected".
- The change was made in the specified lines, as requested
3 In the lines 150 to 152, the sentence should corrected as "For the phenolics and flavonoid content, 0.64 and 5.62 g/L, were detected as quercetin and gallic acid equivalents, respectively.".
- The specified change was made in the manuscript
You should remember that it's not the concentration or content to be detected or accumulated, it should be that the matter with a certain concentration is detected or accumulated. As the same as the other similar errors in this manuscript. Pls carefully check and revise this manuscript.
Thank you for your comment, but unfortunately I did not understand what errors are being pointed at. If the changes you are looking for are language-related, we are also submitting a Grammarly report on english language, hoping that this is sufficient for the manuscript to be published. If not, we are happy to improve the document in a very short period of time. If the nature of the pointed errors is different, let us know and we will make the required adjustments as son as possible.

Reviewer 2 Report
All comments have been taken into account.
Yours faithfully,
Reviewer
Round 2
RESPONSES TO THE EDITOR´S/REVIEWER’S COMMENTS
Manuscript ID: fermentation-2311375
Title: Initial study of fungal bioconversion of guishe (Agave lechu-guilla residue) extract for bioherbicide activity on model seeds
Dear Editor - Fermentation
Please find below the detailed responses to the Reviewer’s/Editor’s comments. We have also enclosed the thoroughly revised version of the manuscript mentioned above according to the Reviewer’s comments, suggestions, and recommendations. The changes made have been highlighted with a yellow background (Round 1 Revisions) in color as per the Editorial’s recommendation.
Reviewer 2
All comments have been taken into account.
- Thank you for your support
